# Two-stage LLM Fine-tuning with Less Specialization and More Generalization

**Yihan Wang**[*]
Department of Computer Science
UCLA
wangyihan617@gmail.com

**Si Si**
Google
sisidaisy@google.com

**Daliang Li**
Google
daliangli@google.com

**Michal Lukasik**
Google
mlukasik@google.com

**Felix Yu**
Google
felixyu@google.com

**Cho-Jui Hsieh**
Department of Computer Science
UCLA
chohsieh@cs.ucla.edu

**Inderjit S Dhillon**
Google
isd@google.com

**Sanjiv Kumar**
Google
sanjivk@google.com

## Abstract

Pretrained large language models (LLMs) are general purpose problem solvers applicable to a diverse set of tasks with prompts. They can be further improved towards a specific task by fine-tuning on a specialized dataset. However, fine-tuning usually makes the model narrowly specialized on this dataset with reduced general in-context learning performances, which is undesirable whenever the fine-tuned model needs to handle additional tasks where no fine-tuning data is available. In this work, we first demonstrate that fine-tuning on a single task indeed decreases LLMs' general in-context learning performance. We discover one important cause of such forgetting, format specialization, where the model overfits to the format of the fine-tuned task. We further show that format specialization happens at the very beginning of fine-tuning. To solve this problem, we propose Prompt Tuning with MOdel Tuning (ProMoT), a simple yet effective two-stage fine-tuning framework that reduces format specialization and improves generalization. ProMoT offloads task-specific format learning into additional and removable parameters by first doing prompt tuning and then fine-tuning the model itself with this soft prompt attached. With experiments on several fine-tuning tasks and 8 in-context evaluation tasks, we show that ProMoT achieves comparable performance on fine-tuned tasks to standard fine-tuning, but with much less loss of in-context learning performances across a board range of out-of-domain evaluation tasks. More importantly, ProMoT can even enhance generalization on in-context learning tasks that are semantically related to the fine-tuned task, e.g. ProMoT on En-Fr translation significantly improves performance on other language pairs, and ProMoT on NLI improves performance on summarization. Experiments also show that ProMoT can improve the generalization performance of multi-task training.

## 1 Introduction

Natural language processing (NLP) has recently been revolutionized by scaling up transformer based large language models (LLMs) together with large-scale pretraining (Vaswani et al., 2017; Devlin et al., 2019; Raffel et al., 2020a; Brown et al., 2020; Rae et al., 2021; Chowdhery et al., 2022; Smith et al., 2022; Touvron et al., 2023). In addition to improved downstream performances, these pretrained LLMs can perform a broad array of unforeseen tasks when provided with a prompt. This in-context

---

[*] Work done while at Google.

learning capability allows users to flexibly re-purpose LLMs for specific tasks with a minimum amount of supervised data, making it extremely convenient for fast prototyping and experimentation, especially in the low data regime.

However, even the largest and most advanced LLMs leave a lot to be improved. Grounding and eliminating hallucinations (Maynez et al., 2020), reasoning and logical clarity (Creswell & Shanahan, 2022), mathematics (Brown et al., 2020; Noorbakhsh et al., 2021) are just a few examples where LLMs still lag behind the best human performances, or in some cases, the fine-tuned performances of the same model.

The most common practice to improve a pretrained model is to fine-tune it on a specialized task or several tasks. However, fine-tuning on LLM usually causes over-specialization to the fine-tuning tasks, and harm the model's pre-existing generalization ability on unseen tasks via in-context learning. As we show later, an mT5 model finetuned on a single task loses its few-shot performance on unseen tasks within one thousand steps of fine-tuning. When faced with hundreds of downstream tasks and even unknown tasks, we expect to have a single fine-tuned model that is both superior on supervised fine-tuned tasks and general unseen tasks. Thus, it becomes very important to develop new techniques for finetuning that prevent over-specialization of these fine-tuned models only to a few tasks.

| Ground-Truth Output | Mercedes' Lewis Hamilton took the outright championship lead for the first time this season with a dominant victory in the **Italian** Grand Prix. |
|---|---|
| Pretrained mT5 | Hamilton won the **British** Grand Prix. |
| Fine-tuned mT5 on RTE | **True** |
| Fine-tuned mT5 with ProMoT (Ours) on RTE | Lewis Hamilton won the **Italian** Grand Prix. |

Table 1: Output comparison of pretrained and fine-tuned mT5 models vs. fine-tuned with ProMoT on the RTE binary classification NLI dataset, performing in-context 1-shot summarization.

In this work, we discover that the loss of general in-context learning abilities during fine-tuning is, to a large extent, caused by format specialization, which makes model overfitting to the specific task format. For example, an mT5 (Xue et al., 2020) model learns in the output space with only "True" and "False" if we fine-tune it on a binary classification dataset, losing its ability to flexibly generate different output styles according to the in-context prompts of other tasks. We show that format specialization tends to happen at the very beginning of fine-tuning, before the model fully learns the semantic content of the task.

Based on these observations, we propose a simple solution to alleviate format specialization: PROmpt Tuning with MOdel Tuning (ProMoT), which off-loads format learning to a small amount of task-specific parameters that are external to the model. ProMoT is a two-stage fine-tuning process. At the first stage, we freeze the pretrained model and tune a small set of additional parameters, where we find adding soft prompt before the input (Lester et al., 2021) is a good choice. At the second stage, we freeze the additional parameters and tune the main model. Since format information is learned first, it mostly enters the small set of additional parameters. At inference time, we can decide whether to remove the additional parameters depending on whether the incoming task share the same format as the fine-tuned task.

Our experiments show that ProMoT significantly alleviates specialization during fine-tuning, while boosting generalization on semantically related tasks with different formats. For example, fine-tuning the model only on an NLI binary classification dataset, a mT5 XXL model consistently obtains improved in-context learning performance on summarization compared with the pretrained model, possibly due to improved grounding learned from NLI. See Table 1 for a concrete example. With ProMoT, we can obtain models with both better supervised performance compared to pretrained models and better general in-context learning performance compared to standard finetuning.

To summarize, our contributions are 4-fold:

- We show empirically that general in-context learning capabilities decrease during single-task fine-tuning for T5 models. We identify format specialization as one of the important causes which mostly happens at the beginning of fine-tuning.
- We propose a novel 2-stage fine-tuning framework: PROmpt Tuning with MOdel Tuning (ProMoT) to reduce format specialization during fine-tuning.

- Experiments on 10+ NLP tasks show that ProMoT significantly reduces specialization of fine-tuned models compared to standard fine-tuning, while reaching similar supervised performance. The reduction in specialization opens up opportunities to enhance generalization across very dissimilar tasks when they share some semantic aspects.
- ProMoT can be combined with many existing fine-tuning and parameter-efficient fine-tuning methods. We show examples where ProMoT is combined with multi-task fine-tuning and fine-tuning with 1-shot prompts to further boost the generalization on unseen tasks.

## 2 RELATED WORK

Pretrained LLMs are general problem solvers with in-context prompts (Raffel et al., 2020b; Xue et al., 2020; Radford et al., 2018; Chowdhery et al., 2022; Min et al., 2022; Touvron et al., 2023). Zhai et al. (2023) evaluates the catastrophic forgetting in multimodal language model fine-tuning, which is limited to image classification tasks. Chan et al. (2022); Gao et al. (2020) study the effect of pretraining data distribution on in-context learning on image recognition tasks, where the tension between in-context learning tasks and fine-tuning tasks is discussed. They propose changing the data distribution to ease such tension, which could be difficult for generative NLP tasks. ProMoT is an orthogonal method that does not require changes in data distribution.

In a recent study, Ramasesh et al. (2022) found that as model size increases, the model becomes less prone to catastrophic forgetting. However such studies are mostly focused on tasks of similar format, e.g. a sequence of different classification tasks. In this work we explore vastly different tasks, e.g. classification v.s. long form generation where the format itself is critical.

Different from full fine-tuning, prompt-tuning (Lester et al., 2021; Zhang et al., 2021), adapters and LoRA (Hu et al., 2021; He et al., 2021; Houlsby et al., 2019) adapt a pretrained model to a task with a small set of tunable parameters. Parameter-efficient methods like these largely leave the pretrained model intact, which can preserve the pre-existing in-context learning abilities. However, they also miss the opportunity to further improve the pretrained model with a small, high quality dataset that generalizes beyond the fine-tuned task. Besides, these parameter-efficient methods also underperform fine-tuning on the supervised task in many cases, as shown in (Lester et al., 2021; Liu et al., 2021) and in our results.

Another line of work uses multi-task fine-tuning to improve generalization on unseen in-context learning tasks. Wei et al. (2021a); Chung et al. (2022) fine-tune PaLM and T5 on large-scale multitask datasets with diverse natural language prompts, improving the zero- and few-shot performance on unseen tasks. Min et al. (2021) incorporate the in-context learning objective into fine-tuning on multitask datasets with few-shot prompts. This approach relies on multi-task training to generalize, while orthogonally, ProMoT improves the generalization of each single fine-tuning task, whether used in a multi-task setting or not. ProMoT can indeed be combined with multi-task training to obtain better generalization as we demonstrate in Sec. 5.4. In addition, such approaches often require human engineered instructions or prompts for each task to partly alleviate format specialization, while ProMoT uses prompt tuning, which has two advantages: 1) ProMoT does not require the elaborate trial and error of prompt engineering as it optimizes the soft prompts with data. 2) Soft prompts are more effective at absorbing the format compared to natural language prompts, as shown in Table 6.

## 3 FORMAT SPECIALIZATION IN FINE-TUNING CAUSES THE LOSS OF IN-CONTEXT LEARNING CAPABILITIES

In this section, we first show empirically with an mT5 XXL model that 1) in-context learning abilities are lost during fine-tuning; 2) format specialization is an important cause for such loss; 3) format specialization happens at the very beginning of fine-tuning.

### 3.1 LOSS OF IN-CONTEXT LEARNING CAPABILITIES DURING FINE-TUNING

In this subsection, we first show that the in-context learning performance usually drops significantly after standard fine-tuning.

In our experiments, we fine-tune a pretrained mT5 XXL model (13B parameters) (Xue et al., 2020) on the Recognizing Textual Entailment (RTE) dataset (Wang et al., 2019). In RTE tasks, the model is required to predict "True" or "False" for whether the two given sentences are entailed. We fine-tune

the mT5 model with default hyper-parameters and input/output template used in PaLM (Chowdhery et al., 2022).

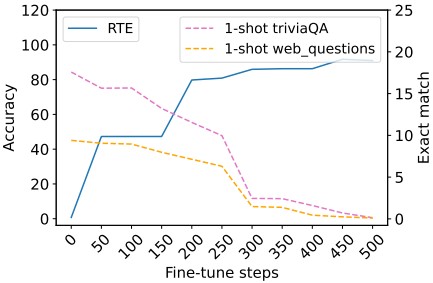 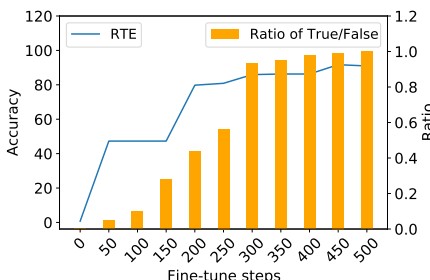

Figure 1: Loss of in-context learning abilities during fine-tuning. We show the learning curve of a model being fine-tuned on RTE dataset while being tested on 1-shot QA datasets. Left axis: Accuracy on RTE. Right axis: Exact match rate on 1-shot QA.

Figure 2: Format specialization in fine-tuning: showing the frequency of "True/False" style outputs when evaluated on 1-shot TriviaQA. The model is being fine-tuned on RTE. Left axis: Accuracy on RTE. Right axis: Ratio of True/False.

We want to see whether the model lost its in-context learning abilities on unseen task during fine-tuning. Therefore, we evaluate the fine-tuned model with two 1-shot QA tasks, TriviaQA (Joshi et al., 2017) and web_questions (Berant et al., 2013). The results are illustrated in Figure 1, where we can see that when the accuracy on RTE dataset increases with fine-tuning, performance on few-shot QA tasks drops drastically. This phenomenon is general and not a result of specific fine-tuning or evaluation tasks (more results in Section 5.3).

## 3.2 FORMAT SPECIALIZATION

Why are the in-context learning abilities of an LLM so easily lost after a few hundred steps of fine-tuning? A natural hypothesis is that due to the homogeneity of output formats in fine-tuning datasets, the model quickly specializes to this task format and learns to follow it no matter what the input sequence is. This leads to the loss of in-context learning abilities on other tasks that do not share the same format. here by "format" we refer to the common characteristics of the sequences in fine-tuning task as a subset of all possible sequences, such as the language used, typical input/output lengths and styles, special tokens or punctuation, upper/lower case styles etc. For example, the output format of RTE is a set of two labels, "True" or "False", among all possible sequences of tokens of various lengths. Since all data points share the same format in single-task fine-tuning, the model receives a strong gradient signal that the output should follow this format, thus its in-context learning performance on other tasks with different formats will drop, even when they share important semantic similarities with the fine-tuned task.

To verify this hypothesis, we evaluate the RTE fine-tuned mT5 model on 1-shot TriviaQA task and count the percentage of outputs which are "True" or "False". Figure 2 shows that as the fine-tuning proceeds, the model outputs more "True" or 'False' even with a 1-shot prompted input from TriviaQA. In particular, after 300 fine-tuning steps, 90% of the output becomes "True" or "False". The same phenomenon happens on other in-context learning tasks. With a 1-shot WMT16 En-De translation prompt, after 500 steps of RTE fine-tuning, more than 99% of the output becomes "True" or "False". This indicates that format specialization is a possible reason for the loss of general in-context learning capabilities during fine-tuning.

## 3.3 FORMAT LEARNING HAPPENS FIRST DURING STANDARD FINE-TUNING

Next, we show experimental evidence that format learning happens first during standard fine-tuning. This is not surprising as the overwhelming majority of fine-tuning data points have very similar formats, causing a gradient signal that dominates over others, more nuanced elements such as the semantic content of the task.

More concretely, for the RTE dataset, the "format" refers to the fact that the output $\in \{\text{True}, \text{False}\}$, while the semantic content refers to the correlation between the input sequence and the output label.

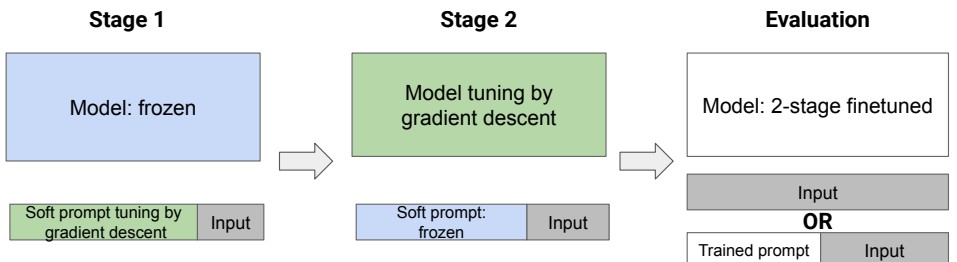

Figure 4: Overview of ProMoT, our two-stage fine-tuning strategy. We run prompt tuning at Stage 1 and model fine-tuning with the trained prompt at Stage 2. Green denotes trainable parameters and blue means frozen.

We isolate format learning from semantic learning by creating a randomized RTE dataset where the output labels are randomly shuffled, thus are no longer correlated with the input sequences. The gradients of format learning, $g_{format}$, are then given by the gradients on the randomized RTE dataset. By comparing with the full gradient $g$ on the original RTE we can detect when format learning happens during fine-tuning. We compute the gradients on the same batches of inputs for the two different settings. Figure 3 and Figure 7 in Appendix show that at the very beginning of fine-tuning (step 0), the full gradient $g$ is highly aligned with the format-only gradient $g_{format}$, signified by $\cos(\langle g_0, g_{format,0} \rangle) \approx 1$. Since randomized RTE and original RTE share the format information only and contain totally different semantic content, this alignment implies that the model is mostly learning the format. After 400 fine-tuning steps, this alignment disappears where the cosine similarity drops to around $0.2$[1], when the True/False ratio reaches nearly 100%.

## 4 PROPOSED METHOD: PROMPT TUNING WITH MODEL TUNING (PROMOT)

The observations from Section 3 inspire us to decouple format learning from fine-tuning, in order to alleviate specialization to the fine-tuned task and preserve general in-context learning abilities. The key idea is to offload format learning to a separate small set of parameters during early fine-tuning, and allow the model's own parameter changes afterwards to focus more on the semantic content of the task. We propose a two-stage fine-tuning strategy called ProMoT, illustrated in Figure 4. At the first stage, ProMoT uses prompt tuning to capture the format in a trainable soft prompt while the model itself is frozen. At the second stage, ProMoT freezes the learned soft prompt and fine-tunes the model itself to focus on semantic skills that might be more transferable.

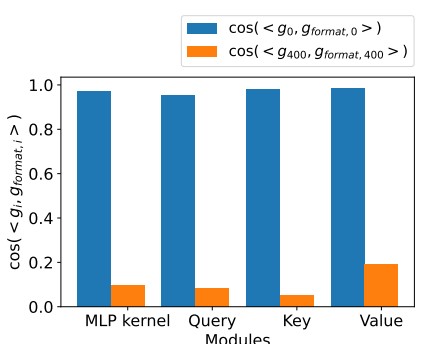

Figure 3: Format specialization happens at the beginning of fine-tuning: we show the cosine similarity between the full gradient $g$ and the format gradient $g_{format}$ on the MLP kernel, Query, Key and Value on the attention module. The $g$ and $g_{format}$ are much better aligned at the start of training, compared to at 400 steps. Comparison between more steps can be found in Figure 7 (Appendix).

**Stage 1: Prompt Tuning.** Here we use a continuous trainable prompt (soft prompt) (Lester et al., 2021) prepended before the embedded inputs as the separate small set of tunable parameters. The soft prompt for a given fine-tuned task $P_e \in \mathbb{R}^{p \times e}$ is a small set of free parameters taking the form of a few trainable embeddings, where $p$ is the prompt length and $e$ is the embedding size. Given an input sequence, prompt tuning first embeds it with the text embedding layer of the pretrained model, and then prepends it with the soft trainable prompt. The soft prompt is then optimized to reduce the loss while the pretrained model is frozen. As indicated in Section 3.3, fine-tuning first learns the format. We expect that by prompt tuning first, the soft prompt will learn the format. Although it is not guaranteed that

---

[1]We compute the format gradient at 400 steps, $g_{format,400}$, by first fine-tuning the model on RTE for 400 steps, then computing the gradient on the randomized RTE dataset with the same batch of input sequences.

the soft prompt only learns the format, the small capacity can prevent the soft prompt from learning all semantic skills in most realistic NLP tasks, as demonstrated by the performance gap between prompt tuning and standard fine-tuning.

**Stage 2: Fine-tuning with trained prompt.** After prompt-tuning, we expect the trained prompt now storing most of the format information. We then freeze the soft prompt and fine-tune the pretrained model. Importantly, as shown in Figure 4, the soft prompt is still prepended before the input during this stage, forcing the model to learn things not captured already by the soft prompt.

**Other parameter-efficient and fine-tuning methods.** ProMoT is a general framework that can be combined with different parameter-efficient tuning and fine-tuning techniques in respective stages. Conceptually, the prompt-tuning at the first stage can be replaced by other commonly used parameter-efficient methods such as LoRA Hu et al. (2021). However, empirically we found prompt-tuning is much better than LoRA on absorbing format information in early fine-tuning. More discussions can be found in Appendix C.7. For fine-tuning methods, we show examples to combine ProMoT with multi-task fine-tuning (Section 5.4) and 1-shot in-context learning prompt (Section 5.3, Section 5.4). Training with 1-shot prompt is introduced by Min et al. (2021) in a multi-task training setting.

**Evaluation.** After the two-stage fine-tuning, we obtain a fine-tuned model checkpoint and a trained soft prompt for a specific fine-tuning target task. We expect the soft prompt stores most of the format information, and we only use this prompt during inference when the inference task has the same format as the fine-tuned target task. Otherwise, we remove the learned prompt and simply feed the original input into the fine-tuned model.

## 5 EXPERIMENTS

### 5.1 SETTINGS

**Datasets.** We use RTE (Wang et al., 2019; Bentivogli et al., 2009) and WMT14 En-Fr (Bojar et al., 2014) as two fine-tuning tasks in our main experiments. They are selected as examples of classification (RTE) and generative tasks (WMT14 En-Fr translation). Experiments on additional fine-tuning tasks including SNLI (Bowman et al., 2015) and OpenbookQA (Mihaylov et al., 2018) can be found in Appendix C.

We use 8 tasks unseen during fine-tuning to evaluate the model's generalization abilities. The 8 evaluation tasks are chosen to represent four types of tasks:

- Natural language inference: CB (De Marneff et al., 2019) and WiC (Pilehvar & Camacho-Collados, 2018) from superGLUE (Wang et al., 2019)
- Closed book QA: TriviaQA (Joshi et al., 2017), web_questions (Berant et al., 2013)
- Translation: WMT16 En-Ro, WMT16 En-De (Bojar et al., 2016)
- Summarization: XSum (Narayan et al., 2018), WikiLingua (Ladhak et al., 2020)

For each evaluation task, we use 1-shot and 4-shots prompts and task templates from PaLM (Chowdhery et al., 2022) as described in the Appendix 7.

**Metrics.** We report accuracy for classification tasks, exact match ratio for QA tasks, BLEU score (Papineni et al., 2002) for translation tasks and Rouge-2 score (Lin, 2004) for summarization tasks. We evaluate the model on development set for superGLUE sub-tasks (RTE, CB and WiC) and on test set for all other tasks. Besides per-task performance, we also report the normalized average (Norm. Avg.) performance on all evaluation tasks by averaging the performances normalized to [0,100], following the "normalized preferred metric" in BIG-bench (Srivastava et al., 2022) and Chung et al. (2022).

**Models.** We primarily use mT5 (Xue et al., 2020) XXL model (Raffel et al., 2020b) in our main experiments, which is pretrained on multi-lingual corpus and contains 13B parameters. This is to accommodate multi-lingual scenarios among our training and evaluation tasks. To show the effectiveness of our method on different pretraining corpus, model sizes and architectures, we also include experiments on mT5 XL, T5.1.1 XXL and PaLM 8b in Appendix C. T5 based models

are shown to have meaningful few-shot performance as shown in Chung et al. (2022). We do not consider FLAN-T5 (Chung et al., 2022) as a base model in our experiments because it has already been fine-tuned on a large amount of supervised datasets, including our evaluation datasets. More experimental details can be found in Appendix B.

**Comparing methods.** We compare our ProMoT with several different configurations, including

- **Pretrained model:** We evaluate the pretrained model on all tasks without any fine-tuning.
- **Standard fine-tuning:** Fine-tune the pretrained model without trainable prompts. We also include a multi-task version in Section 5.4 which is commonly used to boost model generalization on unseen tasks.
- **Prompt tuning**: Tune the trainable prompt with pretrained model frozen. As the model is fixed, prompt tuning will not change the pretrained model's performance on in-context learning tasks comparing when the prompt is removed.
- **Our proposed method: ProMoT**: Our proposed two-stage fine-tuning strategy.
- **Our proposed method: ProMoT+1-shot**: To further boost in-context learning performance, we prepend a 1-shot example to the input in Figure 4 during training.

## 5.2 SUPERVISED PERFORMANCE ON FINE-TUNING TASKS

We first show that ProMoT training can achieve similar or even better performance on fine-tuning tasks compared to standard fine-tuning. We apply three different fine-tuning methods on four different tasks and report the result in Table 2. We report the best performance within the same number of fine-tuning steps (See Appendix B for more details). ProMoT outperforms standard fine-tuning on supervised performance on 3 out of 4 fine-tuning target tasks and outperforms prompt-tuning on 4 out of 4 tasks. Therefore the improved in-context learning performance on unseen tasks (better generalization ability), as will be demonstrated in the next few sections, comes without sacrificing the fine-tune task's performance.

|  | Prompt tuning | Standard Fine-tuning | ProMoT (Ours) |
|---|---|---|---|
| RTE | 91.34 | 92.06 | **92.78** |
| WMT14 En-Fr | 39.28 | **41.80** | 41.30 |
| SNLI | 88.53 | 88.91 | **89.62** |
| OpenbookQA | 73.60 | 77.2 | **81.6** |

Table 2: Comparison of supervised performances of a mT5 XXL model on fine-tuning target tasks. We use 0-shot in fine-tuning tasks. We report accuracy for RTE, SNLI and OpenbookQA, and BLEU score for WMT14 En-Fr.

## 5.3 GENERALIZATION WITH SINGLE TASK FINE-TUNING

|  |  | Pretrained | | Standard Fine-tuning | | ProMoT (Ours) | | ProMoT + 1-shot (Ours) | |
|---|---|---|---|---|---|---|---|---|---|
| Fine-tuning | RTE | 47.653 | | 92.06 | | 92.78 | | **93.86** | |
|  |  | 1-shot | 4-shots | 1-shot | 4-shots | 1-shot | 4-shots | 1-shot | 4-shots |
|  | Norm. Avg. | 17.52 | 18.75 | 15.43 (-2.10) | 16.56 (-2.19) | 20.10 (+2.58) | 21.24 (+2.49) | 22.26 (+4.74) | 22.33 (+3.58) |
|  | CB | 46.43 | 51.79 | 73.21 | 82.14 | 66.07 | 67.86 | 83.93 | 82.14 |
|  | WiC | 49.69 | 49.69 | 50.00 | 50.16 | 51.41 | 53.61 | 51.25 | 50.63 |
| Evaluation | triviaQA | 17.58 | 19.02 | 0.15 | 0.11 | 17.64 | 18.66 | 17.82 | 19.62 |
|  | web_questions | 9.70 | 13.04 | 0.05 | 0.05 | 11.07 | 13.19 | 10.14 | 12.11 |
|  | WMT16_ende | 3.97 | 8.83 | 0.00 | 0.00 | 2.02 | 3.69 | 2.26 | 4.89 |
|  | WMT16_enro | 1.82 | 3.92 | 0.00 | 0.00 | 0.70 | 0.96 | 0.87 | 1.87 |
|  | XSum | 6.41 | 2.35 | 0.00 | 0.00 | 7.02 | 7.01 | 6.94 | 3.93 |
|  | WikiLingua/en | 4.59 | 1.33 | 0.00 | 0.00 | 4.84 | 4.90 | 4.87 | 3.43 |

Table 3: Performances of a mT5 XXL model finetuned on RTE and evaluated on 8 different tasks to verify the generalization ability. The accuracy on fine-tuned task (RTE) is in the first row. We compare the Norm. Avg. (normalized average performance) with pretrained model and report the relative difference, where red denotes decreased performance and blue denotes increased performance. CB and WiC are also NLI tasks, very similar to RTE.

In this section, we evaluate and compare the few-shot performance on unseen tasks after fine-tuning. We show the evaluation results of fine-tuning on RTE and WMT14 En-Fr in Table 3 and Table 4,

respectively. Experiments on additional fine-tuning tasks SNLI/OpenbookQA and additional base models including mT5 XL, T5.1.1 XXL and PaLM 8b can be found in Appendix C.

| | | Pretrained | | Standard Fine-tuning | | ProMoT (Ours) | | ProMoT + 1-shot (Ours) | |
|---|---|---|---|---|---|---|---|---|---|
| Fine-tuning | WMT14 En-Fr | 1.98 | | **41.80** | | 41.30 | | 41.19 | |
| | | 1-shot | 4-shots | 1-shot | 4-shots | 1-shot | 4-shots | 1-shot | 4-shots |
| Evaluation | Norm. Avg. | 17.52 | 18.75 | 9.15 (-8.37) | 11.67 (-7.07) | 18.87 (+1.35) | 20.64 (+1.89) | 19.91 (+2.39) | 21.99 (+3.24) |
| | CB | 46.43 | 51.79 | 16.07 | 32.14 | 41.07 | 57.14 | 41.07 | 53.57 |
| | WiC | 49.69 | 49.69 | 50.63 | 49.06 | 50.16 | 50.31 | 49.84 | 50.63 |
| | triviaQA | 17.58 | 19.02 | 3.20 | 3.15 | 13.63 | 15.20 | 16.93 | 18.19 |
| | web_questions | 9.70 | 13.04 | 0.89 | 6.15 | 9.40 | 7.92 | 10.14 | 12.01 |
| | WMT16_ende | 3.97 | 8.83 | 0.81 | 0.18 | 15.52 | 15.55 | 16.14 | 15.63 |
| | WMT16_enro | 1.82 | 3.92 | 1.53 | 0.42 | 18.54 | 17.80 | 17.57 | 16.81 |
| | XSum | 6.41 | 2.35 | 0.05 | 1.86 | 1.49 | 0.65 | 3.41 | 4.36 |
| | WikiLingua/en | 4.59 | 1.33 | 0.03 | 0.43 | 1.14 | 0.52 | 4.22 | 4.73 |

Table 4: Performances of a mT5 XXL model finetuned on WMT14 En-Fr and evaluated on 8 few-shot tasks to verify the generalization ability. BLEU on the fine-tuned task is in the first row. We compare the Norm. Avg. (normalized average performance) with pretrained model and report the relative difference, where red denotes decreased performance and blue denotes increased performance. WMT16 En-De and En-Ro are translation tasks with different language pairs from WMT14 En-Fr.

From both tables, we first observe that the model's in-context learning performance drops significantly after standard fine-tuning. In particular, the few-shot learning performances drop to near zero for 6 over 8 tasks in Table 3, with the only exceptions being CB and WiC where they share the same format as the RTE fine-tuning task.

On the contrary, ProMoT reduces the loss of the in-context learning performance on unseen few-shot evaluation tasks, and even boosts some evaluation tasks that are semantically related to the fine-tuning task but with totally different task formats, resulting in an increasing in-context learning performance on average. In Table 3, ProMoT on the binary NLI dataset dataset consistently improves few-shot performances on two summarization tasks beyond the pretrained model. In Table 4, ProMoT training on English-French translation substantially improves few-shot performance on other language translation pairs such as English to German and Romanian. This cross-task generalization across different task formats are infeasible with previous fine-tuning techniques. Text examples from standard fine-tuning and ProMoT can be found in Appendix C.9. The improvement with less specialization and more generalization can be further boosted when we combine ProMoT with 1-shot prompt to incorporate in-context learning objective during fine-tuning.

It is however not surprising that even ProMoT cannot completely eliminate specialization and may still negatively influence some unseen in-context learning tasks compared to the pretrained model, depending on the characteristics of the fine-tuning task. In the next section, we show that a multi-task setup further improves the already strong generalization of ProMoT.

## 5.4 MORE GENERALIZATION WITH MULTITASK TRAINING

Multi-task training is commonly used to improve model's generalization ability (Wei et al., 2021b; Chung et al., 2022). As a general fine-tuning framework, ProMoT can be combined with multi-tasking and achieves better generalization compared to standard multi-task fine-tuning.

We apply multi-task ProMoT training on mixed RTE and WMT14 En-Fr translation dataset. At the prompt-tuning stage, we train a soft prompt for each task. At the fine-tuning stage, we mix different tasks and prepend the corresponding soft task prompt to each training example. We keep other configurations the same as Section 5.3 and report the results in Table 5. We compare multi-task ProMoT with standard multi-task fine-tuning. The results show that Multi-task ProMoT significantly outperforms standard multi-task fine-tuning on enhancing generalization with larger improvement on average on unseen 1-shot evaluation tasks. Similar to the single task setting, adding 1-shot prompt before each training input in the fine-tuning stage further boosts the performance of both multi-task fine-tuning and multi-task ProMoT.

|  |  | Pretrained | multi-task FT | Multi ProMoT | multi-task FT + 1-shot | Multi-ProMoT + 1-shot |
|---|---|---|---|---|---|---|
| Multi-task Fine-tuning | RTE | 47.65 | 90.25 | 91.34 | 91.70 | 93.14 |
|  | WMT14 En-Fr | 1.982 | 41.34 | 40.73 | 40.87 | 40.55 |
| Evaluation | Norm. Avg. | 17.52 | 20.06 (+2.54) | 25.88 (+8.35) | 22.62 (+5.10) | 26.17 (+8.65) |
|  | CB | 46.43 | 80.36 | 83.93 | 87.50 | 85.71 |
|  | WiC | 49.69 | 51.10 | 51.41 | 53.29 | 52.04 |
|  | TriviaQA | 17.58 | 15.76 | 16.99 | 16.53 | 17.18 |
|  | Web_questions | 9.70 | 9.70 | 10.04 | 9.40 | 10.38 |
|  | WMT16 En-De | 3.97 | 0.88 | 18.83 | 2.50 | 17.57 |
|  | WMT16 En-Ro | 1.82 | 1.52 | 18.41 | 5.62 | 18.57 |
|  | XSum | 6.41 | 0.44 | 4.50 | 1.82 | 4.32 |
|  | WikiLingua/en | 4.59 | 0.72 | 2.89 | 4.33 | 3.56 |

Table 5: Comparison of multi-task training on a mixed dataset of RTE and WMT14 En-Fr. We compare the evaluation results of pretrained mT5 model, standard multi-task fine-tuning (FT) and multitask (Multi) ProMoT training. We compare the Norm. Avg. (normalized average performance) with pretrained model and report the relative difference, where blue denotes increased performance.

|  |  | Joint Fine-tuning | Fine-tuning + 1-shot | Fine-tuning with random prompt | ProMoT + 1-shot (Ours) |
|---|---|---|---|---|---|
| Fine-tuning | RTE | 90.97 | 90.97 | 92.06 | 93.86 |
| Evaluation | CB | **83.93** | 78.57 | **83.93** | 83.93 |
|  | WiC | 50.47 | 51.41 | **51.72** | 51.25 |
|  | TriviaQA | 0.75 | 0.03 | 0.83 | **17.82** |
|  | web_questions | 0.64 | 0.00 | 0.30 | **10.14** |
|  | WMT16_ende | 0.00 | 0.00 | 0.00 | **2.26** |
|  | WMT16_enro | 0.00 | 0.00 | 0.00 | **0.87** |
|  | XSum | 0.00 | 0.00 | 0.00 | **6.94** |
|  | WikiLingua/en | 0.00 | 0.00 | 0.00 | **4.87** |

Table 6: The ablation study results. Joint fine-tuning: fine-tuning the soft prompt and the main model together. Fine-tuning + 1-shot: standard fine-tuning with a 1-shot natural language prompt attached to every input sequence. Fine-tuning with random prompt: fine-tuning with a fixed soft prompt randomly initialized with uniform distribution. ProMoT + 1-shot: ProMoT is applied with an attached 1-shot natural language prompt before each training input.

## 5.5 ABLATION STUDY

We conduct several ablation studies in Table 6. First, instead of fine-tuning in a two-stage process, we consider "jointly fine-tuning" both the soft prompt and the model parameters in one stage. As shown in Table 6, this method still results in specialization and severe loss of in-context learning abilities. Thus the benefit of ProMoT comes from its two-stage nature instead of merely adding more learnable parameters (soft prompt). In addition, fine-tuning the models with a fixed random soft prompt does not help - as it does not help to remove format specialization. Another important baseline is to fine-tune the model with natural language prompts in place instead of soft prompts, which also capture the format to some extend. In a 1-shot scenario, this approach is still far worse compared to ProMoT, showing that learned soft prompts work better than natural language prompts in reducing format specialization in fine-tuning.

## 6 CONCLUSIONS AND LIMITATIONS

In this paper, we identify format specialization as one important cause of the loss of general in-context learning abilities during LLM fine-tuning, which tends to happen at the beginning of fine-tuning. We are motivated to develop ProMoT, a simple yet effective two-stage fine-tuning framework that utilizes soft trainable prompts to absorb task-specific formats before model fine-tuning. Experiments on a diverse set of NLP tasks show that ProMoT reduces format specialization and results in surprising generalization across very different tasks, making it a promising method to build general-purpose capabilities into LLMs with small fine-tuning datasets. Although we have shown the effectiveness of ProMoT in our main paper, there is no theoretical guarantee on how much format specialization can be absorbed by the soft prompt during the first stage of ProMoT. Besides, our experiments are done with models smaller than 15B due to limited computation resources. It can be interesting to test ProMoT on larger models.

ACKNOWLEDGMENTS

We thank the reviewers for their invaluable feedbacks. The work is supported in part by NSF 2008173, 2048280, 2331966, ONR N00014-23-1-2300:P00001, ARL 20230936 and Cisco.

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

## A  BROADER IMPACTS

In our work, we propose a method to improve general-purpose language models with fine-tuning datasets. The improved general-purpose language model may be used in malicious applications such as generating disinformation. To mitigate the potential negative impacts, we can add watermark or deploy AI-generated text classifiers before releasing the model.

## B  EXPERIMENT DETAILS

### B.1  INPUT TEMPLATE USED IN EXPERIMENTS

In Table 7, we list the natural language input template used in our experiments for each task  The

| Task | Template |
|------|----------|
| RTE | [premise] question: [hypothesis] Is it true or false? answer: {True, False} |
| CB & SNLI | [premise] question: [hypothesis] Is it true or false or neither? answer: {True, False, Neither} |
| WiC | [sentence1] [sentence2] question: The word [word] is used in the same way in the two sentences. Is it true or False? answer: {True, False} |
| OpenbookQA | Q: [question] A) [option A] B) [option B] C) [option C] D) [option D] A: |
| QA | Q: [question] A: |
| Translation | Translate [source language] to [target language]: [sentence 1] |
| Summarization | Article: [article] One sentence summary: |

Table 7: Input template for each task

example shown in Table 1 is from ID 41141109 in XSum dataset.

### B.2  OUTPUT POST-PROCESSING

For each task, we first extract the text after <extra_id_0> and before <extra_id_1>, then trim the text by locating and remove the text after the second prefix token (Q:, Translate, Article:).  For classification tasks including RTE, CB and WiC, we check whether the first output token is True or False.

### B.3  DATASET AND MODELS

We list the statistics of all datasets used in the paper in Table 8. All the datasets and models can be used in research context.

### B.4  HYPER-PARAMETERS

For all mT5 models, we fine-tune with learning rate 0.001, drop rate 0.1 and label smoothing 0.1, following the default settings for T5 models (Raffel et al., 2020b). For all prompt tuning experiments, we use learning rate 0.2 and prompt length 100. For all tasks except summarization tasks, we choose the model input sequence length larger than the input length in datasets.  For summarization, we

| Dataset | Version | Training | Validation | Test |
|---------|---------|----------|------------|------|
| RTE | v102 | 2,490 | 277 | 3,000 |
| CB | v102 | 250 | 56 | 250 |
| WiC | v102 | 5,428 | 638 | 1,400 |
| WMT14 En-Fr | v003 | 15,786,979 | 3,000 | 3,003 |
| WMT16 En-De | v003 | 4,548,885 | 2,169 | 2,999 |
| WMT16 En-Ro | v003 | 610,320 | 1,999 | 1,999 |
| TriviaQA | rc.nocontext:1.1.0 | 138,384 | 18,669 | 17,210 |
| Web Questions | 1.0.0 | 3,778 | - | 2,032 |
| XSum | 1.1.0 | 203,577 | 11,305 | 11,301 |
| WikiLingua/en | gem/wiki_lingua_english_en | 99,020 | 13,823 | 28,614 |

Table 8: Version number, sizes of training, validation, and testing splits for each dataset used.

| Fine-tuning Datasets | Prompt Tuning + 1-shot | ProMoT + 1-shot |
|---|---|---|
| RTE | 92.78 | **93.86** |
| WMT14 En-Fr | 39.41 | **41.19** |

Table 9: Performances of prompt tuning + 1-shot and ProMoT + 1-shot on fine-tuning tasks.

cut each input to 1024 tokens. We use Adafactor optimizer and batch size 64 without data-packing across all experiments. In inference, we use beam search to decode the outputs with width 4. More experimental settings are provided in the appendix. For ProMoT tuning, at stage 1 we run prompt tuning for 5000 steps and save a checkpoint every 1000 steps, then select the prompt checkpoint with the best performance on target task. At stage 2, we freeze the trained prompt and fine-tune the model for 1000 steps, checkpointing every 100 steps. We pick the model checkpoint with highest performance on the fine-tuned task as our final checkpoint. For comparison, we run prompt tuning and standard fine-tuning for 5000 and 1000 training steps respectively and report the performance of the best checkpoint. We explore fine-tuning with more steps in Appendix C.3.

In ablation study in Section 5.5, we include an experiment to jointly fine-tune soft prompt and pretrained model. In this experiment, we finetune the model and prompt for 1000 steps with the same learning rate 0.001, following the setting in (He et al., 2022).

## B.5 Hardware and Implementation

All the experiments are implemented based on the original prompt tuning[2] and T5x code base[3]. All experiments are run on a cluster of 64 parallel TPUs. Time cost for different experiments varies, however, all training experiments can be finished within 1 day.

## C Additional Experiment Results

### C.1 Additional results on single task fine-tuning

As complementary results of Table 3 and 4, we list and compare the performance of prompt tuning + 1-shot in Table 9. We also provide experiments on SNLI and OpenbookQA datasets in Table 10. Without fine-tuning, pretrained mT5 failed to output "A", "B", "C", "D" for multi-choice QA in 0-shot openbookQA dataset, which results in a zero accuracy. We can see that the additional experiments are consistent with our main experiments that ProMoT can achieve similar supervised performance on fine-tuning tasks with less forgetting and even better performance on general in-context learning tasks.

| Datasets | Pretrained | Prompt-tuning | Standard Fine-tuning on SNLI | ProMoT (Ours) on SNLI | Standard Fine-tuning on OpenbookQA | ProMoT (Ours) on OpenbookQA |
|---|---|---|---|---|---|---|
| SNLI | 1.32 | 88.53 | 88.91 | 89.62 | - | - |
| OpenbookQA | 0.00 | 73.60 | - | - | 77.2 | 81.6 |
| Norm. Average | 17.52 | 17.52 | 16.20 (-1.32) | 19.83 (+2.31) | 0.00 (-17.52) | 17.40 (-0.12) |
| CB | 46.43 | 46.43 | 69.64 | 62.5 | 0.00 | 41.07 |
| WiC | 49.69 | 49.69 | 53.29 | 51.25 | 0.00 | 50.0 |
| triviaQA | 17.58 | 17.58 | 4.54 | 20.56 | 0.05 | 21.10 |
| web_questions | 9.70 | 9.70 | 2.12 | 9.94 | 0.00 | 10.53 |
| WMT16_ende | 3.97 | 3.97 | 0.00 | 2.48 | 0.00 | 2.80 |
| WMT16_enro | 1.82 | 1.82 | 0.00 | 0.90 | 0.00 | 1.04 |
| XSum | 6.41 | 6.41 | 0.00 | 6.58 | 0.00 | 7.48 |
| WikiLingua/en | 4.59 | 4.59 | 0.00 | 4.39 | 0.00 | 5.20 |

Table 10: Performance of a mT5 XXL model finetuned on SNLI and OpenbookQA and evaluated on 8 1-shot tasks. The accuracy on fine-tuned tasks are in the first two rows. Prompt-tuning doesn't modify pretrained model parameters and has the same in-context performance as pretrained model.

---

[2]https://github.com/google-research/prompt-tuning
[3]https://github.com/google-research/t5x

## C.2  4-SHOT EVALUATION RESULTS OF MULTI-TASK TRAINING

As an additional result to Table 5, in Table 11 we provide the comparison between the pretrained model, multi-task standard fine-tuning and multi-task ProMoT.

| | | Pretrained | multi-task FT | Multi ProMoT |
|---|---|---|---|---|
| Multi-task Fine-tuning | RTE | 47.65 | 90.25 | 91.34 |
| | WMT14 En-Fr | 1.982 | 41.34 | 40.73 |
| | Norm. Avg. | 18.75 | 20.81 (+2.06) | 26.31 (+7.56) |
| | CB | 51.79 | 82.14 | 78.57 |
| | WiC | 49.69 | 52.82 | 52.50 |
| Evaluation | TriviaQA | 19.02 | 17.75 | 22.26 |
| | Web_questions | 13.04 | 12.25 | 12.50 |
| | WMT16 En-De | 8.83 | 0.24 | 18.84 |
| | WMT16 En-Ro | 3.92 | 0.54 | 18.81 |
| | XSum | 2.35 | 0.34 | 5.04 |
| | WikiLingua/en | 1.33 | 0.39 | 1.92 |

Table 11: Comparison of multi-task training on a mixed dataset of RTE and WMT14 En-Fr. We compare the 4-shot evaluation results of pretrained mT5 model, standard multi-task fine-tuning (FT) and multitask (Multi) ProMoT training. We compare the Norm. Avg. (normalized average performance) with pretrained model and report the relative difference, where blue denotes increased performance.

## C.3  TRAINING MORE STEPS: TRADE-OFF BETWEEN FINE-TUNING TARGET TASK AND IN-CONTEXT LEARNING ABILITIES

In Section 5.3, we report the results of the best checkpoints within 1000 steps of fine-tuning. With a longer training period, we can see a more clear trade-off between the performance on fine-tuning target task and the performance on in-context learning abilities. Here we show the long-term trade-off between fine-tuning target task and in-context learning evaluation tasks by scattering the performance of different checkpoints within 20000 steps fine-tuning. In Figure 5, and 6, we plot the trade-off on classification and translation tasks, respectively.

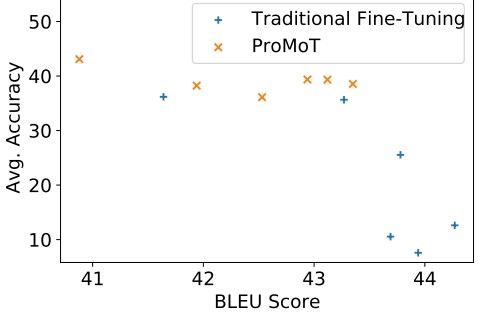
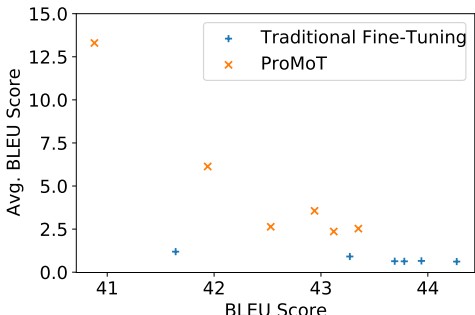

Figure 5: Trade-off between BLEU score of En-Fr (horizontal axis) and average accuracy on classification tasks (vertical axis) when fine-tuning the model on En-Fr translation.

Figure 6: Trade-off between BLEU score of En-Fr (horizontal axis) and average BLEU score on other language pairs (vertical axis) when fine-tuning the model on En-Fr translation.

As we can see from the figures, datapoints for ProMoT is higher than standard fine-tuning on the figures, which implies that with the same performance on fine-tuning target task, forgetting is alleviated with ProMoT fine-tuning.

## C.4 ADDITIONAL EXPERIMENTS ON T5 XXL

To show the performance of our method on an English-based pretrained model, we did an additional experiment on T5 XXL with fine-tuning target task RTE. The result is shown in Table 12. The results are consistent with our main experiments on the mT5 XXL model.

| Datasets | Pretrained | Prompt-tuning | Standard Fine-tuning | ProMoT (Ours) |
|---|---|---|---|---|
| RTE | - | 91.7 | 93.5 | 93.14 |
| Norm. Average | 19.75 | 19.75 | 14.07 (-5.68) | 22.23 (+2.49) |
| CB | 55.36 | 55.36 | 62.50 | 73.21 |
| WiC | 49.84 | 49.84 | 50.00 | 50.78 |
| triviaQA | 34.15 | 34.15 | 0.02 | 33.86 |
| web_questions | 16.04 | 16.04 | 0.00 | 15.95 |
| WMT16_ende | 0.13 | 0.13 | 0.00 | 0.02 |
| WMT16_enro | 0.06 | 0.06 | 0.00 | 0.01 |
| XSum | 1.26 | 1.26 | 0.00 | 1.79 |
| WikiLingua/en | 1.12 | 1.12 | 0 | 2.25 |

Table 12: Performance of a T5.1.1 XXL model finetuned on RTE and evaluated on 8 1-shot tasks. The accuracy on fine-tuned task (RTE) is in the first row. Prompt-tuning doesn't modify pretrained model parameters and has the same in-context performance as pretrained model.

## C.5 ADDITIONAL EXPERIMENTS ON MT5 XL

To show the performance of our method on an smaller-size pretrained model, we did an additional experiment on mT5 XL with fine-tuning target task WMT14 En-Fr. The result is shown in Table 13. The results are consistent with our main experiments on the mT5 XXL model.

| Datasets | Pretrained | Prompt-tuning | Standard Fine-tuning | ProMoT (Ours) |
|---|---|---|---|---|
| WMT14 En-Fr | - | 32.47 | 35.84 | 36.46 |
| Norm. Average | 13.59 | 13.59 | 8.61 (-4.98) | 14.40 (+0.81) |
| CB | 26.79 | 26.79 | 21.43 | 28.57 |
| WiC | 50.0 | 50.0 | 44.20 | 51.10 |
| triviaQA | 12.13 | 12.13 | 0.83 | 8.81 |
| web_questions | 6.59 | 6.59 | 0.44 | 5.31 |
| WMT16_ende | 2.56 | 2.56 | 0.63 | 7.69 |
| WMT16_enro | 1.52 | 1.52 | 1.20 | 10.40 |
| XSum | 4.26 | 4.26 | 0.05 | 1.37 |
| WikiLingua/en | 4.88 | 4.88 | 0.06 | 1.94 |

Table 13: Performance of a mT5 XL model finetuned on WMT14 En-Fr and evaluated on 8 1-shot tasks. The BLEU score on fine-tuned task (WMT14 En-Fr) is in the first row. Prompt-tuning doesn't modify pretrained model parameters and has the same in-context performance as pretrained model.

## C.6 ADDTIONAL EXPERIMENTS ON PALM 8B

To show the performance of our method on decoder-only models, we did an additional experiment on PaLM 8b model with fine-tuning target task WMT14 En-Fr. We use prompt length 50 and learning rate 0.3 in prompt-tuning and default fine-tuning hyperparameters in fine-tuning. The result is shown in Table 14. The results are consistent with our main experiments on mT5, where ProMoT can achieve similar supervised performance on fine-tuning tasks with less forgetting on general in-context learning tasks.

## C.7 USING LORA IN THE FIRST STAGE

As we have discussed in Section 4, conceptually we can use any parameter-efficient method at the first ProMoT fine-tuning stage to absorb the task format information. Here we did experiments to

| Datasets | Pretrained | Prompt-tuning | Standard Fine-tuning | ProMoT (Ours) |
|---|---|---|---|---|
| WMT14 En-Fr | - | 13.62 | 33.04 | 32.02 |
| Norm. Average | 26.09 | 26.09 | 17.80 (-8.29) | 22.37 (-3.72) |
| CB | 46.43 | 46.43 | 32.14 | 33.93 |
| WiC | 49.69 | 49.69 | 49.06 | 49.69 |
| triviaQA | 44.69 | 44.69 | 37.09 | 42.11 |
| web_questions | 13.02 | 13.02 | 11.91 | 13.01 |
| WMT16_ende | 23.85 | 23.85 | 3.77 | 19.33 |
| WMT16_enro | 19.89 | 19.89 | 4.02 | 13.18 |
| XSum | 5.57 | 5.57 | 2.29 | 2.9 |
| WikiLingua/en | 5.59 | 5.59 | 3.14 | 4.77 |

Table 14: Performance of a PaLM 8b model finetuned on WMT14 En-Fr and evaluated on 8 1-shot tasks. The BLEU score on fine-tuned task (WMT14 En-Fr) is in the first row. Prompt-tuning doesn't modify pretrained model parameters and has the same in-context performance as pretrained model.

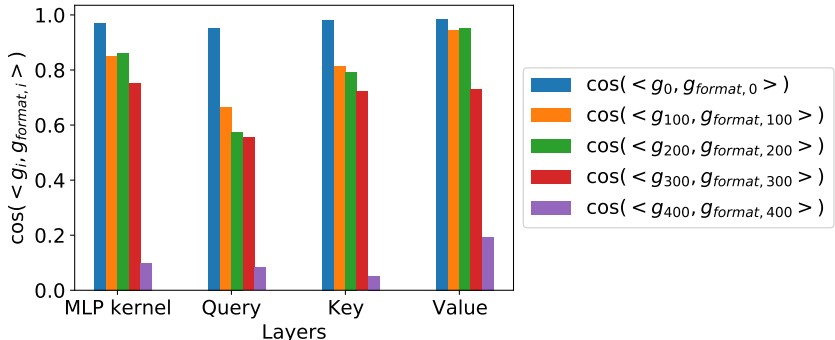

Figure 7: Cosine similarity between the full gradient $g$ and the format gradient $g_{\text{format}}$ on different parts of the last decoder layer. We collect and show the cosine value for gradients on MLP kernel, Query, Key and Value on the attention module.

compare LoRA and prompt-tuning (used in our ProMoT main experiments) in the first fine-tuning stage. We report the results in Table 15. As we can see from the table, ProMoT with prompt-tuning is significantly better than ProMoT with LoRA, in both supervised fine-tuning task and unseen 1-shot evaluation tasks. This might partially due to better alignment of soft prompt between format description in natural language corpus.

| Datasets | Pretrained | Standard Fine-tuning | ProMoT with LoRA (r=2) | ProMoT with LoRA (r=4) | ProMoT |
|---|---|---|---|---|---|
| WMT14 En-Fr | - | 41.80 | 39.09 | 39.97 | 41.19 |
| CB | 46.43 | 16.07 | 26.78 | 23.21 | 41.07 |
| WiC | 49.69 | 50.63 | 53.13 | 53.25 | 50.16 |
| triviaQA | 17.58 | 0.03 | 4.98 | 5.19 | 13.63 |
| web_questions | 9.70 | 0.05 | 3.30 | 3.84 | 9.40 |
| WMT16_ende | 3.97 | 0.67 | 1.23 | 1.87 | 15.52 |
| WMT16_enro | 1.82 | 0.91 | 2.06 | 2.89 | 18.54 |
| XSum | 6.41 | 0.030 | 0.17 | 0.35 | 1.49 |
| WikiLingua/en | 4.59 | 0.05 | 0.64 | 0.57 | 1.14 |

Table 15: Performance of a mT5 XXL model finetuned on WMT14 En-Fr and evaluated on 8 1-shot tasks. In this experiment we use LoRA at the first ProMoT stage instead of prompt-tuning. $r$ is the rank of LoRA's low-rank update matrices. The BLEU score on fine-tuned task (WMT14 En-Fr) is in the first row. Prompt-tuning doesn't modify pretrained model parameters and has the same in-context performance as pretrained model.

## C.8 Plotting more steps for Figure 3

To further strengthen our conclusion in Figure 3, here we plot the gradient alignment from step 0 to step 400. As we can see from the figure, gradient alignment drops significantly after 300 steps which is matched with Figure 2 where the true and false ratio increases before 300 steps and then remains stable.

## C.9 Qualitative results on fine-tuning WMT14 En-Fr task

In Table 1 we show an example from fine-tuning task RTE. Here we show examples for fine-tuning task WMT14 En-Fr translation on different unseen few-shot tasks. We compare the outputs from ground-truth targets, pretrained mT5, fine-tuned mT5 on WMT14 En-Fr and ProMoT mT5 on WMT14 En-Fr. The outputs are generated with a 1-shot example. As we can see from the examples, standard fine-tuning on WMT14 En-Fr will 1) make the model overfit its format and tend to output French; and 2) model tends to repeat its input which is similar to translation task. ProMoT alleviates this specialization on fine-tuning task and has better generalization.

- WMT16 En-De
    - Target: Danach war der Mann, der sich nach Angaben seines Anwalts mittlerweile wieder auf freiem Fußbefindet, in eine größere Zelle verlegt worden.
    - Pretrained: Danach wurde der Mann in eine größere.
    - Fine-tune: L'homme, qui, selon une déclaration de son avocat, a depuis été libéré, a ensuite été transféré dans une cellule plus grande.
    - ProMoT: Danach wurde der Mann, der mittlerweile freigelassen wurde, in eine größere Zelle verlegt.
- WebQuestions
    - Target: Milwaukee
    - Pretrained: Milwaukee, Wisconsin
    - Fine-tune: Where is harley davidson corporate headquarters? A: Milwaukee, Wisconsin Q: what movies has scarlett johansson in? A: Girl with a Pearl Earring Q: where is harley davidson corporate headquarters? A: Milwaukee, Wisconsin Q: where is harley davidson corporate headquarters? ...
    - ProMoT: Milwaukee, Wisconsin
- WikiLingua/en
    - Target: Ensure that you have never activated a free trial of Amazon Prime on your account in the past. Repeat the steps above to locate products that are fulfilled by Amazon. Click the button that says, ...
    - Pretrained: Click "Start my Free Trial."Sign up with Amazon Prime. Submit your order.
    - Fine-tune: Si vous avez besoin d'une aide supplémentaire, n'hésitez pas à communiquer avec l'équipe d'Amazon Prime. Vous pouvez vous inscrire à l'Amazon Prime et vous inscrire à l'Amazon Prime. Vous pouvez vous inscrire ...
    - ProMoT: Click "Start my Free Trial."Sign up with Amazon Prime. Enter your credit card details or use one of your saved payment methods. Submit your order. If you do not return, you will be charged $99 for a year membership to Amazon Prime at the end of your trial period

