# OpenReview forum: "Two-stage LLM Fine-tuning with Less Specialization and More Generalization"
_ICLR.cc/2024/Conference — ICLR 2024 poster_

### Official Review · Reviewer_3Pj7 · 2023-10-31

**Soundness:** 2 fair
**Presentation:** 3 good
**Contribution:** 3 good
**Rating:** 8
**Confidence:** 4

**Summary:**

This work introduces PROmpt Tuning with MOdel Tuning (ProMoT) to address the issue of “format specialization”. This problem occurs when Pre-trained Language Models (PLMs) lose their in-context learning abilities after being fine-tuned for a specific task. To elaborate, PLMs are initially fine-tuned for the task using soft prompt tuning, wherein only the soft prompt is tuned for that specific task. Then, all parameters, excluding the trained soft prompt, are fine-tuned for the same task. Experimental results show that ProMoT not only prevents the format specialziation but also enhances the performance on the target task.

**Strengths:**

- **Reasonable problem setup and simple approach;** The proposed method, ProMoT, is demonstrated to be effective in preventing format specialization using a straightforward two-stage fine-tuning approach. Additionally, this paper reveals the cause of format specialization by analyzing the changes in gradient similarity between the data from both the randomized and original RTE datasets.
- **Notable Experimental Results;** The authors conducted extensive experiments across diverse types of tasks including Natural Language Inference (NLI), closed book QA, translation, and summarization. Furthermore, experimental results show that the proposed method is effective in alleviating the issue of format specialization after being fine-tuned on the classification (RTE) and translation (WMT14 En-Fr) tasks.
- **Clear Presentation;** This paper is well-written. Therefore, it is easy to understand the problem setup and the proposed method.

**Weaknesses:**

- **Limited Evaluation;** While the experimental results include diverse evaluation tasks, the fine-tuning datasets are limited to just two tasks: NLI and Trnaslation. Why did the authors limit their experimentation to only these two tasks? Is it implied that format specialization does not occur in other tasks, such as question answering and summarization? I couldn’t find a satisfactory explanation in the paper regarding the absence of experimental results from fine-tuning on other tasks.

**Questions:**

See Weaknesses; Why do the authors use only two datasets for fine-tuning?

---

> ### Author Response · Authors · 2023-11-19
>
> We thank the reviewer for valuable suggestions and comments. We address your questions below in detail.
>
> **Regarding experiments on finetuning on more tasks.** Sorry for the confusion. In Table 10 of the appendix, we have shown experimental results on two additional finetuning datasets including openbookQA, which is a QA dataset and SNLI as another NLI dataset. Due to space limit, we put these results in Appendix,. The results are consistent with our main experiments in the paper. Also based on our gradient analysis given in section 3.2-3.3, we can see that format specialization happens for finetuning tasks with any well-defined output syntax. The reason is that the gradient associated with such syntax is present consistently in all finetuning data points, resulting in a strong gradient signal. Lots of  fine tuning tasks in the NLP community today contain such syntax elements.
>
> We would deeply appreciate it if the reviewer finds that these clarifications are helpful and strengthens our manuscript, and raise the score accordingly.

---

> ### Author Response · Authors · 2023-11-21
>
> We were wondering if there are any further questions or concerns we could clarify before the discussion period closes tomorrow. We appreciate the feedback, and will appreciate it if you would like to comment on our rebuttal and additional results and revise the score accordingly. Thank you!

---

> ### Comment · Reviewer_3Pj7 · 2023-11-23
>
> Thank you to the authors for their rebuttal. I hope the authors will consider including additional experiments in the main paper in their next revision. After reading comments from other reviewers, my confidence in the quality of the work has increased, leading me to believe that it is sufficiently robust for acceptance. Consequently, I have raised my score to 8.

---

### Official Review · Reviewer_5e4F · 2023-11-01

**Soundness:** 3 good
**Presentation:** 3 good
**Contribution:** 3 good
**Rating:** 6
**Confidence:** 4

**Summary:**

The work focuses on solving the issue of performance decrease after applying single-task finetuning for T5 and PaLM models. The paper proposes a two-stage finetuning: 1) model frozen with soft prompt tuning 2) model tuning with soft prompt tuning frozen. They conduct experiments on 10+ NLP tasks, and the proposed approach achieves similar supervised performance. The paper explores a variety of experimental settings, single-task, and multi-task finetuning, and provides comprehensive insights showing the generalization ability of the model.

**Strengths:**

- The proposed method is shown to be very effective in preserving the model generalization with single-task fine-tuning while it slightly reduces the specialization, which is very beneficial for LLMs.
- Comprehensive evaluation on multiple tasks, and the authors run multiple runs on the few-shot setting to better show the model performance variance.
- The experimental setting comprises many interesting settings (single-task vs. multi-task).
- It is a well-written paper with good clarity on the technical experimental details.

**Weaknesses:**

- The limitation of the paper is only evaluated on encoder-decoder models. It would be interesting to also evaluate the approach on decoder-only models.

**Questions:**

- Do you have any difference trends on different base models? and also different model types (e.g., decoder-only models)?

---

> ### Author Response · Authors · 2023-11-19
>
> We thank the reviewer for valuable suggestions and comments. We address your questions below in detail.
>
> **Regarding experiments on more models (different base models including decoder-only model).** We have done additional experiments on different base models beyond mT5 shown in the main paper, including T5XXL, T5XL and PaLM-8B. Due to limited space in the main paper, we put these results in Table 11, 12, 13 in Appendix. The PaLM-8B model we used in the additional single-task experiments is a decoder-only model. We report the results in Table 13 in the Appendix. As we can see from these results, the additional results are consistent with our main experiments on the mT5 XXL model, where ProMoT outperforms prompt-tuning in-domain and outperforms standard finetuning out-of-domain. In particular, as an improved finetuning method, ProMoT shows significantly less forgetting of in-context capabilities compared to standard finetuning and may even improve over out-of-domain but semantically related tasks compared to the pretrained model.

---

### Official Review · Reviewer_qDAR · 2023-11-01

**Soundness:** 2 fair
**Presentation:** 3 good
**Contribution:** 3 good
**Rating:** 8
**Confidence:** 4

**Summary:**

This paper proposes an innovative fine-tuning method that preserves the model's general in-context learning ability. The method involves absorbing task-specific format knowledge into prompt tuning parameters and finetuning the main model. During evaluation, the prompt tuning parameters can be optionally dropped depending on dataset similarity.

**Strengths:**

1. The method is "strange". It furthers our understanding of param-efficient fine-tuning methods by showing and leveraging a unique property of prompt tuning.
2. Evaluated on a diverse set of datasets.
3. Ablation experiments showing the necessity of some design details.

**Weaknesses:**

1. Only on one model.
2; Finetuned on only two datasets.

**Questions:**

1. In Table 3,4, It may be worthwhile to put more highlights on the most relevant datasets, e.g. CB and WiC for RTE, WMT16 en>de, en>ro for WMT14 en>fr.

---

> ### Author Response · Authors · 2023-11-19
>
> We thank the reviewer for valuable suggestions and comments. We address your questions below in detail.
>
> **Regarding experiments on more models.** Sorry for the confusion. We have done additional experiments on other models beyond mT5, including T5XXL, T5XL and PaLM-8B. Due to limited space in the main paper, we put these results in Table 11, 12, 13 in Appendix. As we can see from these tables, results on these additional models are consistent with our main experiments on the mT5 XXL model, where ProMoT outperforms prompt-tuning in-domain and outperforms standard finetuning out-of-domain. In particular, as an improved finetuning method, ProMoT shows significantly less forgetting of in-context capabilities compared to standard finetuning and may even improve over out-of-domain but semantically related tasks compared to the pretrained model.
>
> **Regarding experiments on more datasets.** We have included experiments on two additional finetuning datasets (SNLI and OpenBookQA) in Appendix Table 10 for the mT5 XXL model. The results are consistent with our main experiments in the paper.
>
> **Regarding highlighting similarity tasks.** Thanks for the suggestions. We have put some texts in the tables’ captions to highlight the improvement over the similar tasks.

---

### Official Review · Reviewer_ptkK · 2023-11-04

**Soundness:** 2 fair
**Presentation:** 2 fair
**Contribution:** 2 fair
**Rating:** 5
**Confidence:** 3

**Summary:**

The paper proposes to use a two-step finetuning approach to reduce forgetting when the finetuned model is applied to other tasks. This is done by prompt tuning the model for some number of steps while freezing the rest of the model and then freezing the prompt while tuning the rest of the model in the subsequent steps. The hope is that the initial prompt tuning learns the format and the rest of the tuning learns the task. The performance is evaluates on a set of NLP tasks where the authors show competitive on task performance and better out-of-task performance.

**Strengths:**

- The idea to separate format learning and task learning is interesting.
- The paper is easy to follow.
- The experiments show improvements in out-of-task performance.

**Weaknesses:**

- I am not sure if there's practical utility to the proposed setting. Improving transfer to other tasks which are data limited could be a use case but I found the multi-task section of the paper very underwhelming. Multi-task training was done on only 2 tasks while it would be more informative to do it on a lot more tasks, results over multi-task 1-shot baseline are very mixed and is also missing 4-shot comparison.
- Prefix + Parameter tuning baseline seems to be missing. What happens if you do both prefix and parameter tuning but don't use the prefix embedding for other tasks. This should be done in both single task and multi-task setting.
- It would be good to include a prefix-tuning column. This would have some performance for the source task and same performance as the pre-trained model on other tasks.
- It is hoped that the prefix-tuning stage learns the format. It would be good to empirically test this hypothesis. One simple experiment is to check if the prefix embedding outputs the source task format on other tasks.
- Results on Palm 8B in the appendix don't look as promising and they should not be in the appendix but in the main paper. The approach seems to show worse performance than pre-trained model on other tasks after using the proposed approach. Would also be good to include 4-shot results here.

**Questions:**

See weaknesses.

---

> ### Author Response · Authors · 2023-11-19
>
> We thank the reviewer for valuable suggestions and comments. We address your questions below in detail with additional experiments and in-depth analysis.
>
> **Regarding practical utility of ProMoT.** Thanks for the comments. We agree with the reviewer’s insight that the most significant application of ProMoT would be in domains where high quality finetuning data is limited. In such scenarios, it is imperative to have methods that are 1) robust against severe forgetting due to finetuning data homogeneity, which will cause format specialization as we extensively demonstrated in this work.
> 2) can maximize the value of each datapoint through improved generalization.
> We would like to stress that this kind of situation is not a small niche, especially when language models are applied to new and complex domains where it is extremely difficult to collect high quality supervised finetuning data even for one task - autonomous LLM agents is one such area amongst many. Given this successful proof of concept, we believe that our method will open up new possibilities in these domains and will be of interest to a broad community of ICLR researchers.
>
> **Regarding massive multi-task experiments.**  Thanks for the comments. As discussed above, the focus of this work is more towards low data / high generalization requirement regime. With our best effort given computational resources constraints, we choose not to run massive multi-task training on large-scale datasets (such as the FLAN mixture, among others). However, as demonstrated by a recent work (Table 1 in https://openreview.net/pdf?id=VrHiF2hsrm), forgetting of in-context learning capabilities also exists in the massive multi-task regime, where methods like ProMoT could be helpful. We hope to explore the massive multi-task regime with ProMoT as a future work, but we do believe that our current results are self-contained, thorough and significant enough to be known by the community.
>
> **Regarding 4-shot evaluation results.** Thanks for the suggestions. We have added the 4-shot evaluation results for fine-tuning v.s. ProMoT with multi-task training in the table below. As we can see from the table, the results are consistent with the 1-shot result that ProMoT can produce a generally better model on these evaluation tasks with 4-shots prompts. Note that as True/False format has been seen during fine-tuning, the model with standard fine-tuning can also do well on CB and WiC tasks. We have added this additional result to the Appendix.
>
> |               | Pretrained Model - 4-shots | Multi-task Finetune - 4-shots | Multi-task ProMoT - 4-shots |
> |---------------|----------------------------|-------------------------------|-----------------------------|
> | cb            | 51.79                      | 82.14                         | 78.57                       |
> | wic           | 49.69                      | 52.82                         | 52.5                        |
> | TriviaQA      | 19.02                      | 17.75                         | 22.26                       |
> | Web_questions | 13.04                      | 12.25                         | 12.5                        |
> | WMT16 En-De   | 8.83                       | 0.24                          | 18.84                       |
> | WMT16 En-Ro   | 3.92                       | 0.54                          | 18.81                       |
> | XSum          | 2.35                       | 0.34                          | 5.04                        |
> | WikiLingua/en | 1.33                       | 0.39                          | 1.92                        |
>
> **Regarding the Prefix + Parameter tuning baseline.** Thanks for the comments. We have included a ‘joint fine-tuning’ setting in the ablation study (Table 6), which trains the soft prompt and the main model jointly exactly as the comments suggested. In evaluation, we only use the main model and remove the soft prompt on evaluation tasks with different formats. As we can see from the table, the prefix + parameter tuning joint training still suffers from significant loss of general in-context learning performance, as the format learning can not be separated into the prompt if we train the two parts jointly.
>
> **Regarding the prefix-tuning column.** Thank you for your suggestion on the table organization. For the source tasks, we report the in-domain performance of prompt-tuning in Table 2, showing that prompt-tuning is indeed inferior compared to standard finetuning or ProMoT in-domain. For evaluation on other tasks, exactly as the reviewer mentioned, since prompt tuning does not touch the parameters of the pretrained model, it has the same few-shot performance on evaluation tasks as the pretrain model where we remove the soft prompt. We have also made this clear to the audience in the footnotes of Table 3 and 4. We hope these would clarify the prompt tuning results well.

---

> ### Author Response · Authors · 2023-11-19
>
> **Regarding evaluating prefix-tuning on other tasks.** Thanks for the suggestions. As requested by the reviewer, we evaluated the performance of using the prompt trained from the English to French translation task WMT14 En-Fr using prompt tuning on other translation tasks with different language pairs. We show some examples below, we add a 1-shot prompt before the actual input in evaluation denoted by [1-shot prompt].
>
> As we can see from the table, when we use the prompt from WMT14 En-Fr to test on En-De and En-Ro translation tasks, the predictions are still in French, which implies that the format information of outputting in French is learned by the prompt.
>
> | Input                                                                                                                                                                                                                      | Target                                                                                                                                                                            | Prediction with the soft prompt trained from En-Fr            |
> |----------------------------------------------------------------------------------------------------------------------------------------------------------------------------------------------------------------------------|-----------------------------------------------------------------------------------------------------------------------------------------------------------------------------------|---------------------------------------------------------------|
> | [1-shot example omitted…]   Translate English to German: The first series of The Lenny Henry Show aired in 1984, and in the Nineties he was known as, among other things, chef Gareth Blacklock in the comedy series Chef! | Die erste Reihe der Lenny Henry Show ging 1984 auf Sendung, und in den Neunzigern war er unter anderem als der K\u00fcchenchef Gareth Blacklock in der Comedyserie Chef! bekannt. | Nous souhaitons aider nos clients \u00e0 r\u00e9ussir.        |
> | [1-shot example omitted…]  Translate English to German: I know Georgy won't go.                                                                                                                                            | Ich wei\u00df, dass Georgij nicht hingehen wird.                                                                                                                                  | Cela ne signifie pas une Europe minimaliste \u2013 et je veux |
>
>
> | Input                                                                                                                             | Target                                                                                                                      | Prediction with the soft prompt trained from En-Fr                                                    |
> |-----------------------------------------------------------------------------------------------------------------------------------|-----------------------------------------------------------------------------------------------------------------------------|-------------------------------------------------------------------|
> | [1-shot example omitted…]  Translate English to Romanian: People using ad-blocking software are likely to run into a few hitches. |  Persoanele care folosesc programe software de blocare a reclamelor \u00eent\u00e2mpin\u0103 c\u00e2teva constr\u00e2ngeri. | Les personnes qui utilisent des logiciels de blocage des annonces |
> | [1-shot example omitted…] Translate English to Romanian: Brent futures were up 61 cents Wednesday at $48.36 per barrel.           | Cot\u0103rile Brent au crescut miercuri cu 61 de cen\u021bi, ajung\u00e2nd la 48,36 USD per baril.                          | Les futures sur le Brent ont gagn\u00e9 61 centimes mercredi      |
>
>
> **Regarding the performance on PaLM.** Thanks for the comments. We would like to highlight that ProMoT, as an improved finetuning method, still vastly outperformed prompt-tuning in-domain and significantly out-performed standard finetuning out-of-domain, providing a very attractive trade-off point between the two extremes. However, we acknowledge that our results on PaLM models are somewhat more exploratory, which is why they are placed in the appendix. Factors such as the effectiveness of prompt tuning on these models will affect their ability to absorb the format part of the finetuning task. This is an interesting topic that warrants a dedicated examination in a future work.
>
> We would deeply appreciate it if the reviewer finds that these additional experiments and clarification are helpful and strengthens our manuscript, and raise the score accordingly.

---

> ### Author Response · Authors · 2023-11-21
>
> We were wondering if there are any further questions or concerns we could clarify before the discussion period closes tomorrow. We appreciate the feedback, and will appreciate it if you would like to read our rebuttal and additional results, and revise the score accordingly. Thank you!

---

### Author Response · Authors · 2023-11-23

Dear reviewers,

We have performed additional experiments and added more analysis and discussion to address every concern from every reviewer. We would deeply appreciate it if the reviewers find these efforts valuable.

We were wondering if there are any further questions or concerns we could clarify before the discussion period closes today. We appreciate the feedback, and will appreciate it if you would like to read our rebuttal and additional results, and revise the score accordingly. Thank you!

Paper 8041 authors

---

### Meta-Review · Area_Chair_3zp8 · 2023-12-16

**Metareview:**

The submission tackles the problem where fine tuning a model on a task can hurt its performance on other tasks. The paper argues this is partly due to the model specializing its answer format for specific tasks, hurting generalization. To solve it, the authors propose a two-stage finetuning procedure, in which first trainable prompt is learned with respect to a frozen model, and then the remaining parameters are fine-tuned. The method is shown to allow strong performance on the tuning task without hurting generalization. Reviewers thought the paper was well written and the method shows good results. The major criticism was the relatively limited experiments in the original submission, which has been somewhat strengthened by including more tasks in the revision. The additional complexity of the procedure may also limit how much it is used in practice. Overall, I think the paper is borderline, but leaning positive.

**Justification For Why Not Higher Score:**

The additional complexity of the procedure may also limit how much it is used in practice.

**Justification For Why Not Lower Score:**

The method might well be useful for some people.

---

### Decision · Program_Chairs · 2024-01-16

Accept (poster)